# A 20-Year Retrospective Analysis of Plant Poisoning Cases at the Naval Hospital, Varna, Bulgaria

**DOI:** 10.3390/toxins17040197

**Published:** 2025-04-12

**Authors:** Stanila Stoeva-Grigorova, Maya Radeva-Ilieva, Stela Dragomanova, Gabriela Kehayova, Simeonka Dimitrova, Simeon Marinov, Petko Marinov, Marieta Yovcheva, Diana Ivanova, Snezha Zlateva

**Affiliations:** 1Department of Pharmacology, Toxicology and Pharmacotherapy, Faculty of Pharmacy, Medical University of Varna, Varna 9002, Bulgaria; maya.radeva@mu-varna.bg (M.R.-I.); stela.dragomanova@mu-varna.bg (S.D.); gabriela.kehayova@mu-varna.bg (G.K.); simeonka.dimitrova@mu-varna.bg (S.D.); petko.marinov@mu-varna.bg (P.M.); snezha.zlateva@mu-varna.bg (S.Z.); 2Department of Urology, Faculty of Medicine, Medical University of Varna, Varna 9002, Bulgaria; dr.marinov.simeon95@gmail.com; 3Clinical Toxicology Department, Naval Hospital, Varna 9000, Bulgaria; marietyovcheva@gmail.com; 4Department of Biochemistry, Molecular Medicine and Nutrigenomics, Faculty of Pharmacy, Medical University of Varna, Varna 9002, Bulgaria; divanova@mu-varna.bg

**Keywords:** intoxications, plants, toxidromes, anticholinergic toxidrome, cyanogen toxidrome, ricin toxidrome

## Abstract

The nature and epidemiology of plant intoxications are still not well understood, with recent data being limited. The present study aims to report cases of plant poisoning in the clinical practice of the Clinical Toxicology Department at the Naval Hospital—Varna, Bulgaria, over a 20-year period (2003–2023). A documentary retrospective analysis of the hospitalized cases of poisoning with poisonous plants and their grouping into toxidromes was performed. During the study period, patients with plant poisoning admitted to our hospital unit accounted for 0.35% of a total of 12,857 hospitalized individuals. The distribution across the toxidromes based on clinical presentation revealed the highest frequency of anticholinergic, cyanogen, and ricin toxidromes. The majority of the intoxications resulted from unintentional exposure to plant toxins in adult individuals. Most cases followed a mild to severe clinical course, with patient discharge occurring between 2 and 5 days. No fatalities were recorded, thanks to the reported treatment methods. A relatively low incidence of plant-related poisonings was observed, with their predominant manifestations affecting the gastrointestinal, nervous, and cardiovascular systems. Increased reporting of epidemiological data and clinical experiences in the management of plant intoxications would substantially enhance researchers’ understanding of them and facilitate the development of a standardized treatment protocol.

## 1. Introduction

Plant intoxications represent a significant medical issue. The American Association of Poison Control Centers (AAPCC) reports that nearly 8% (50,000) of all annual incoming calls are directly linked to plant exposure [1]. According to the American 41st Annual Report of the National Poison Data System, plants continue to be among the substances most frequently involved in human exposures, including those in adult, pediatric, and pregnant patients [2]. These incidents may arise either unintentionally, deliberately through the improper use of plants and plant-based products, or intentionally for the purpose of self-harm [3,4]. Unintentional exposure through oral ingestion and to external body parts constitutes the most common case, regardless of age groups and social categories [5,6]. This fact is used by some authors to explain the low mortality associated with this type of intoxication [1]. This is observed particularly frequently in children, who constitute the most vulnerable group of individuals due to their innate desire for environmental exploration and relatively underdeveloped immune system [5,7,8]. According to data from the Wikitox information system, 85% of all poison center calls related to plants involve children under the age of six [9]. Unintentional exposures follow and occur by improper usage or, less frequently, suicide attempts through the ingestion of plants or their extracts [4,10]. In recent decades, increasing experimentation with hallucinogenic plants among adolescents has been observed. These detrimental practices, alongside the intentional ingestion of plants for suicidal purposes, manifest with significantly greater severity and underpin the increasingly rapid progression of clinical cases of plant intoxication [11]. An additional risk factor contributing to this increased frequency of exposure is the growing popularity and accessibility of ornamental plants, whose toxic potentials are often inadequately or entirely unexplored [1].

Fatal outcomes from plant intoxications are rare. However, it is important to emphasize that the progression and severity of intoxication are strongly influenced by multiple factors:○*Factors related to the plant*—the plant species, the quantity and duration of exposure, the specific plant part involved, the extent to which the plant material has been chewed or mechanically fragmented prior to ingestion, etc.;○*Factors related to the patient*—age, route of administration, comorbidities, genetic predisposition, enzymatic profile, concurrent medication use, etc. [5,12].

It is also important to note that a single plant may contain several toxic compounds. The most common biologically active compounds capable of inducing a clinical presentation of poisoning are presented in Table 1 [13,14,15]. Plants contain a multitude of xenobiotics that may exert their effects both independently and synergistically. Moreover, certain plant families may share similar or even identical toxins, while others harbor compounds that remain unidentified to date [1].

Despite a few reports on plant poisonings, the nature and epidemiology of such intoxications remain insufficiently studied, with recent data being scarce [10,11,12,16,17,18,19,20,21,22,23]. In comparison to medicinal toxicology, the study of plant-derived xenobiotics is a relatively underdeveloped field. Within this complex medico-toxicological context, a standardized protocol for the specific treatment of plant poisoning is still lacking. Most cases are managed symptomatically, with antidotes used only in rare cases [10]. Therefore, we propose that a more frequent reporting of epidemiological data and clinical experiences in managing plant intoxication would significantly aid in the development of a standardized treatment protocol.

The present study aims to report cases of plant poisoning encountered in the clinical practice of the Clinical Toxicology Department at the Naval Hospital in Varna, Bulgaria, over a span of 20 years (2003–2023). It aims to provide valuable insights to readers regarding the local prevalence of plant poisoning and may serve as a basis for comparison with reports from other regions globally.

## 2. Results

During the 20-year study period (2003–2023), patients with plant poisoning admitted to our hospital unit accounted for 0.35% of the total hospitalized population. This represented 45 patients out of a total of 12,857 hospitalized individuals. Of all the cases, thirty-five were adults with a mean age of 46.46 years (SD 15.62, range 20–75), while ten were minors, with a mean age of 9.7 years (SD 3.13, range 6–16, two children ≤ 6 years old). In over 70% of the cases, the intoxications occurred in female patients. In six cases, the exposure was intentional. Clinical symptoms played a pivotal role in diagnosing and managing the patients admitted to the hospital unit with intoxication. In the majority of the plant intoxication cases, a mild to severe clinical course was observed. Hospitalization typically lasted between 2 and 5 days. No fatal outcomes were recorded during the 20-year reporting period. The toxic syndromes identified are summarized in the following subsections.

### 2.1. Anticholinergic Syndrome (Tropane Alkaloids)

Sixteen patients were registered with anticholinergic syndrome (nine males and seven females). A total of fifteen cases were attributed to the consumption of *Datura stramonium* L. seeds, while one case resulted from the ingestion of an *Atropa belladonna* decoction. Half of the cases involved children, aged between 6 and 14 years, residing in foster care, who had consumed food contaminated with *Datura stramonium* seeds. Three adults deliberately ingested the seeds for recreational purposes to achieve hallucinogenic effects, while one case involved a 58-year-old man who was the victim of an attempted murder by poisoning, carried out by his wife.

*Datura stramonium* L. and *Atropa belladonna* L. contain a cocktail of biologically active substances that block muscarinic cholinergic receptors, including atropine, scopolamine, hyoscyamine, and hyoscine, as well as saponins and gastroenterotoxins [24,25]. In their combined sixteen cases, the symptoms observed in the intoxicated individuals were moderate to severe. The symptoms included dry mouth (xerostomia), throat discomfort, blurred vision, mydriasis, psychomotor agitation, occasional hallucinations, dry and reddened skin on the face and neck, dry mucous membranes, sinus tachycardia, and urinary retention. Treatment was initiated with an antidote (galantamine), gastric lavage with activated charcoal, infusions of electrolytes and furosemide, B vitamins, benzodiazepines, haloperidol, and piracetam. As a result, all patients were discharged in clinically healthy condition after 2–5 days.

Other plants containing these toxins, or those with similar effects on the human body, include *Cestrum nocturnum* L., *Cestrum diurnum* L., *Hyoscyamus niger* L., *Lycium barbarum* L., and *Mandragora officinarum* L. [26,27,28].

### 2.2. Cyanogenic Syndrome

Nine female patients with a history of consuming bitter apricot (*Prunus armeniaca* L.) and/or almond nuts (*Prunus amygdalus* L.), as well as one male patient who consumed the unripe fruits of a black elder (*Sambucus nigra* L.), sought treatment at the clinic. All their intoxications resulted from unintentional exposure. Mild to moderate symptoms were observed, including nausea, fear, tachypnea, dyspnea, sinus tachycardia, and mild metabolic acidosis. Diagnosis was based on the anamnestic data regarding the consumption of bitter nuts and the clinical presentation. Treatment for cyanogenic syndrome included gastric lavage with activated charcoal, supplemental oxygen, vitamin C, and cyanocobalamin (Vitamin B12). All patients were discharged in clinically healthy condition after 2–3 days.

The toxins amygdalin, (S)-sambunigrin, linamarin, and cycasin are cyanogenic glycosides that release hydrogen cyanide upon complete hydrolysis. These glycosides are found in a wide range of plant species, with approximately 2500 known. The plant species most important to the population in Bulgaria include the apricots, peaches, pears, apples, and plums from the Rosaceae family. Plants and their seeds contain amygdalin, which is metabolized into cyanide. The most poisonous cyanogenic species in North America is cassava (*Manihot esculenta* Crantz), which contains linamarin [29,30,31]. Several antidotes are known for cyanogenic glucosides, namely sodium thiosulphate, amyl nitrite, and hydroxocobalamin [31].

### 2.3. Ricin Syndrome (Toxalbumins)

During the study period, seven patients who ingested castor beans (*Ricinus communis* L.) were registered. Six of them consumed the beans orally to treat constipation, while one did so after misidentifying them as peanuts. The toxicities presented with moderate to severe symptoms. The clinical picture included abdominal pain, nausea, vomiting, diarrhea, and dyselectrolytemia. Treatment included gastric lavage with activated charcoal, forced diuresis, antispasmodics, metoclopramide, pantoprazole, and other symptomatic agents. All patients were discharged in clinically healthy condition after 2–5 days.

In Bulgaria, the castor plant is grown ornamentally. It is also found in the wild and is popular for its laxative effect. The toxin in the plant is called toxalbumin because of its proteinaceous nature. Chemically, it is a lectin whose intracellular delivery and, consequently, its toxic action require a series of steps: binding via the ricin B chain to various cell surface glycolipids or glycoproteins containing β-1,4-linked galactose residues; internalization into the cell through endocytosis; entry of the toxin into early endosomes, followed by its transport to the trans-Golgi network via vesicular trafficking; retrograde vesicular transport through the Golgi apparatus to the endoplasmic reticulum; reduction of the disulfide bond linking the ricin A chain to the B chain; partial unfolding of the A chain, rendering it competent for translocation across the endoplasmic reticulum membrane via the Sec61p translocon; partial avoidance of ubiquitination, which would lead to rapid degradation by cytosolic proteasomes immediately after membrane translocation while still partially unfolded; refolding into its protease-resistant, biologically active conformation; and interaction with the ribosome to catalyze the depurination reaction [1,32,33,34]. Clinical symptoms include pronounced gastrointestinal manifestations such as vomiting and profuse diarrhea with dehydration and dyselectrolytemia. The organs affected are the heart, bone marrow, liver, and kidneys [33,34].

It is worth noting the case of a 17-year-old boy who sought telephone consultation after intentionally ingesting a cluster of white acacia (*Robinia pseudoacacia* L.) flowers for experimental purposes. His symptoms were mild, leading him to decline hospitalization, which was one of our predefined exclusion criteria. Consequently, his data were not included in the present study. White acacia is a common tree species in Bulgaria. In addition to its pleasant aroma during blooming, its clusters have a sweet taste that attracts some individuals. The toxic lectins it produces have a ricin-like effect, and clinically, these intoxications are most often mild to moderately severe. Other similar toxic lectins include abrin (from *Abrus precatorius* L., wild pea), phoratoxin and ligatoxin (from certain mistletoe species, *Phoradendron* spp.), and cystatin (from wisteria plants, *Wisteria* spp.), which primarily cause enterocolitis. Lectins are potent cytotoxins and are being explored as potential biological weapons [35,36].

### 2.4. Effect on Na^+^ Channels

During the study period, four patients who mistakenly ingested an oral decoction of white hellebore (*Veratrum album* L.) prepared for the treatment of hair loss were registered. They sought treatment with complaints of fatigue, dizziness, nausea, vomiting, arterial hypotension, and bradycardia. Rhythm and conduction disorders were detected on the ECG, including sinus bradycardia, ventricular and supraventricular extrasystoles, first-degree AV block, complete AV block, and ST changes in the ischemic type. Treatment included gastric lavage with activated charcoal, infusions of moderate amounts of electrolytes, a diuretic, atropine, sodium bicarbonate, magnesium chloride, pantoprazole, dopamine, and group B vitamins. All patients were discharged with normal blood pressure and a restored sinus rhythm. They were discharged clinically healthy after 2–3 days.

White hellebore is a popular herb in Bulgaria that contains the alkaloid veratridine and is primarily used to treat hair loss. Poisonings are accidental and often result from dietary mistakes, as the root resembles leek (*Allium ampeloprasum* L.), and the aerial parts resemble gentian (*Gentiana lutea* L.), both of which are used for teas and wines in Europe. The mechanism of action of hellebore toxins involves sodium channel opening, but with a shorter duration than aconitine. Although toxicity can be severe, treatment with infusions, atropine, and dopamine is usually successful, and deaths are rare [29]. Other plants that contain toxins acting on sodium channels that toxicologists should be aware of include *Aconitum* spp., *Delphinium* spp., *Veratrum* spp., *Rhododendron* spp., *Azalea* spp., *Kalmia* spp., and *Leucothoe* spp. [37].

### 2.5. Cardiovascular Syndrome (Digoxin Syndrome)

During the observed period, three patients were registered at our Toxicology Clinics who mistakenly ingested an alcoholic extract of oleander (*Nerium oleander* L.) leaves. They presented with symptoms such as abdominal pain, nausea, and bradycardia. Sinus bradycardia and a characteristic depression of the ST segment were noted on the ECG. Treatment included gastric lavage with activated charcoal, moderate infusions of electrolytes, and atropine. No specific antibodies or digitalis Fab fragments were administered. After 72 h, the patients were discharged with a normal ECG, stable hemodynamics, and in a state of clinical health.

Poisoning with almost any cardioactive steroid is clinically indistinguishable from digoxin (*Digitalis lanata* Ehrh.) poisoning. The toxicity of plant cardioactive steroids is different from that of digoxin tablets because of the longer plasma half-life of plant toxins, which is 168 h to 192 h, compared to 30–40 h for pharmaceutical digoxin [38]. Other plants that contain heart-active steroids and may cause a similar clinical picture include the following:–Apocynaceae: *Nerium oleander* L., *Strophanthus* spp., *Cascabela thevetia* (L.) Lippold;–Asclepiadaceae: *Asclepias* spp., *Calotropis* spp., *Euonymus europaeus* L.;–Brassicaceae: *Cheiranthus* spp., *Erysimum* spp.;–Liliaceae: *Convallaria majalis* L., *Drimia maritima* (L.) Stearn;–Ranunculaceae: *Helleborus odorus* L.;–Scrophulariaceae: *Digitalis purpurea* L., *Digitalis lanata* Ehrh [39,40].

### 2.6. Nicotine Syndrome

Two patients were diagnosed with nicotine syndrome. The first was a woman who had consumed a salad containing wild parsley (*Aethusa cynapium* L.). Symptoms of poisoning included nausea, vomiting, abdominal pain, diarrhea, mild tremors of the limbs, headache, dizziness, and muscle weakness, with preserved consciousness and moderate agitation. Sinus tachycardia with a frequency of 110 beats per minute and elevated arterial pressure were recorded. Treatment included gastric lavage with activated charcoal, infusions of electrolyte solutions, beta-blockers, and diazepam. After 2 days, the patient was discharged in clinically healthy condition. Wild parsley resembles *Conium maculatum* L. (poison hemlock), which contains the toxin coniine. The most famous case of coniine poisoning is that of Socrates, who was condemned to death by a cup of poisoned hemlock in 399 BC [41]. It stimulates the nicotinic cholinergic receptors (N-receptors), first increasing the entry of acetylcholine, and then suppressing it, stopping the transmission of the neuromuscular impulse. Clinical symptoms of excited nicotinic receptors include salivation, nausea, vomiting, incontinence, hypertension, and tachycardia. In the late paralytic phase, the paralysis of the intercostal muscles and respiratory depression are observed [42]. They exhibit a combination of excitatory sympathetic and M-cholinomimetic effects, including salivation, nausea, vomiting, incontinence of the pelvic reservoirs, hypertension, and changes in heart rate, which may present as either bradycardia or tachycardia. Miosis or mydriasis may also be observed. The most characteristic symptoms are myofibrillations of the chest and limbs, as well as convulsive activity. In the late paralytic phase, respiratory suppression occurs due to paralysis of the intercostal muscles [41,43].

Tobacco (*Nicotiana tabacum*), primarily recognized for its abuse potential through inhalation (containing nicotine), can also induce poisoning via ingestion (chewing tobacco) or transdermal absorption, particularly among workers engaged in tobacco harvesting. After acute exposure, symptoms generally manifest in a biphasic progression. The initial phase is characterized by nicotinic cholinergic stimulation, presenting with abdominal pain, hypertension, tachycardia, and tremors. This is followed by a delayed inhibitory phase, marked by hypotension, bradycardia, and dyspnea, which may ultimately progress to coma and respiratory failure [44]. In this context, a case was documented involving a 10-year-old child with a diagnosed intellectual disability (oligophrenia) who ingested a cigarette containing tobacco. Gastric lavage with activated charcoal was performed 30 min after the incident. The child was monitored for the subsequent 48 h, during which no excitatory syndromes developed, and no further pharmacological intervention was required.

Other toxins that activate nicotinic receptors are lobeline, sparteine, N-methylcytosine, cytisine, and coniine. They are found either independently or simultaneously in the following plant species: *Nicotiana tabacum* L., *Conium maculatum* L., *Lobelia inflata* L., *Laburnum anagyroides* Medik., *Cytisus scoparius* (L.) Link, *Lupinus latifolius* Lindl. ex J.Agardh, *Caulophyllum thalictroides* (L.) Michx., *Sarracenia* spp., and *Aloe* spp. [41,43,45].

### 2.7. Oxalic Acid and Oxalic Crystals

One patient was registered who ingested approximately 250 mL of grated root of *Alocasia x amazonica* (Araceae) with suicidal intent. The patient presented with both local and systemic toxic symptoms, including corrosive-necrotic changes in the oral cavity, esophagus, and intestine, accompanied by severe throat pain, abdominal pain, hematemesis (approximately 400–500 mL), and bloody diarrhea. The patient’s overall condition was impaired, with consciousness suppressed to the point of somnolence. Pulmonary interstitial edema developed within the first hours, subsequently progressing to alveolar edema. Transient hematuria was also recorded. Laboratory findings revealed the presence of calcium oxalates in the urine, metabolic acidosis, and slightly elevated transaminases. Treatment focused on managing both local and systemic toxic manifestations.

Plants of the genus *Alocasia* are widely popular and cultivated as ornamental species. They contain insoluble calcium oxalates in the foliage. Oxalic acid is the strongest acid among the carboxylic acids found in living organisms and sparingly forms soluble chelates with calcium and other divalent cations. The insoluble calcium oxalate crystals (raphides), found in certain plants typically from the *Araceae* family, are associated with a protein toxin, which intensifies the painful irritation of the skin or mucous membranes. However, in this specific case, the exposure was oral through an extract of grated plant root, which contained both oxalic acid and soluble calcium oxalates. Upon ingestion, a rapid onset of symptoms, including redness, swelling, and local pain in the mouth and throat, was observed. This immediate manifestation of symptoms generally limits further exposure [46,47,48]. Unfortunately, in this particular case, the exposure was intentional and for self-harm, which led to the disregard of this “warning” until the complete consumption of the prepared extract. Eye contact causes intense pain, chemical conjunctivitis, and corneal abrasion. The oxalic acid and soluble calcium oxalates present in plant extracts cause metabolic acidosis and damage multiple organ systems, including the kidneys, nervous system, lungs, cardiovascular system, urogenital system, and liver. Calcium oxalates precipitate and deposit in the capillaries and vessels of these organs [49]. Oxalate poisoning may also occur following exposure to plant families such as *Araceae*, *Chenopodiaceae*, *Polygonaceae*, *Amaranthaceae*, and several grass families [48,50,51].

### 2.8. Gastrointestinal Syndrome

A 67-year-old woman was registered who had been consuming water prepared from the skins of old potatoes (*Solanum tuberosum* L.) for one month as a treatment for diabetes mellitus. The patient complained of nausea and dizziness. The skins and sprouts of old potatoes contain the toxic alkaloid solanine, which may induce gastrointestinal symptoms (nausea, vomiting, and diarrhea), electrolyte imbalances, and abdominal pain. These symptoms typically manifest 2–24 h after ingestion and may persist for several days. However, our patient was discharged home without the need for monitoring or supportive therapy, justified by the observed low toxicity, the elapsed time since exposure, and the absence of toxicity during medical observation.

The highest concentration of solanine is found in green potatoes and their tops, but it is also present in many plants of the Solanaceae family:–*Lycopersicon* spp. (Solanaceae) (green tomato)—contains solanine and chaconine alkaloid;–Ranunculaceae family—contains protoanemonin;–*Solanum americanum* Mill. (Solanaceae)—contains the alkaloids solasodine, soladulcidine, solanine, and chaconine with gastrointestinal, neurological, and weak anticholinergic effects;–*Solanum dulcamara* L. (Solanaceae)—contains the alkaloids solanine, chaconine, and atropine with gastrointestinal, neurological, and some anticholinergic effects;–*Solanum nigrum* L. (Solanaceae)—contains solanine, chaconine, and atropine [52,53,54].

### 2.9. Local Skin Syndrome

A 54-year-old woman who works with plants was registered and treated for a generalized erythematous rash on her body. The skin on her hands and lower limbs was swollen, and small bullae containing clear fluid formed in some areas. The skin on her hands and feet, which were exposed to intense sunlight and were not protected by clothing, showed signs of irritation. A diagnosis of phytophotodermatitis was made. Treatment included systemic corticosteroids, antihistamines, and topical corticosteroids. The symptoms showed regression after 7 days. One month after discharge, pigmentation persisted on the skin of her lower extremities. It is suspected that the observed symptoms resulted from exposure to oxalic acid and oxalate crystals, as these are common causes of local mechanical injury and the onset of allergic reactions or phytophotodermatitis. Topical and systemic corticosteroids remain the primary treatment options. Plants containing oxalic acid have been previously listed [9,29].

To date, our medical unit has not encountered any poisonings associated with cholinomimetic syndrome, hepatic carcinogenicity, seizure syndrome (cicutoxin), glycyrrhizin syndrome, or teratogenicity. The frequency distribution of the individual cases of plant intoxication over the studied twenty-year period is presented in Figure 1. Additionally, Table 2 provides a summarized overview of the distribution of patients by the observed severity of toxicity.

## 3. Discussion

A variety of chemical, immunoenzymatic, and DNA-based analyses have been developed to effectively aid in the identification of toxic plants for diagnostic purposes [5,6,29]. The recognition of a specific type of species is also greatly facilitated thanks to the wealth of information on Internet platforms such as WikiTox, POISINDEX, PLANTO, Goldfrank toxicological manuals, etc. [9,29,30,55]. Some authors even acknowledge the utility of artificial intelligence applications with visual recognition capabilities [56]. However, despite their high specificity, many of these methods are not routinely employed in clinical practice or, even when implemented, frequently fail to yield results with the requisite speed for prompt clinical decision-making. In fact, clinical toxicologists have adapted to provide emergency assistance to intoxicated individuals through the rapid and accurate assessment of medical manifestations, as they largely reflect the type of toxin and the mechanism of toxic action. In this way, although through an indirect approach, the diagnosis may be made in a timely manner, from which therapeutic decisions may subsequently arise. In addition, the report by Harchelroad et al. (1988) is particularly intriguing, as it established that emergency department medical personnel correctly identified only 13% of toxic plants [57]. Investigating whether the availability of modern analytical techniques would improve these results would be of interest. However, similar to the majority of reported cases, in our retrospective study, the diagnosis of plant poisoning was primarily based on the history of contact with a specific plant species and clinical presentation, while laboratory findings played a lesser role. Another prerequisite for the success of this strategy is that commonly used antidotes operate on the principle of functional antagonism [58]. This is likely one of the reasons why, to date, the standardization of recommendations for managing such intoxications has been delayed. As it turns out, the treatment of plant intoxications primarily relies on individual case reports, expert opinions, and clinical experimental studies with a small number of voluntary participants, which are more the exception. As expected, due to the nature of plant poisonings, randomized controlled trials in this field are lacking [5].

The present retrospective report acknowledges that cases of accidental or intentional poisoning with plants over the 20-year period observed by our medical unit are significantly rarer compared to incidents involving pharmaceutical, alcoholic, dietary, and other toxic exposures. These cases accounted for 0.35% of the total hospitalized population (12,857). The majority of plant poisoning cases occurred in women, which we attribute to a combination of biological and behavioral factors [11,22,59]. At the same time, a literature review revealed an overall exposure frequency among all patients that was often more than ten to fifteen times higher [7,11,60,61]. We ascribe this discrepancy in the data to differences in the nature of the institutions to which intoxication reports are directed, as well as to local demographic and infrastructural factors. The frequency of plant exposition cases presented in our study is closer to those reported by Davanzo et al. (2011) (2% of the total number), Martínez Monseny et al. (2015) (1% of the total number), and Gummin et al. (2024) (2.02%) [2,21,22]. On the other hand, when the limited epidemiological data are analyzed, it becomes evident that the majority of exposure cases do not lead to contact with toxicology services or hospitals, thereby complicating the accurate assessment of the true prevalence of plant poisoning. For instance, more than 80% of the exposure reports to plant toxins submitted to the AAPCC were recorded as asymptomatic, less than 20% developed mild to moderate symptoms, and fewer than 7% required medical assistance [1]. Clearly, such poison control centers are alerted to a much broader range of exposed individuals (including cases that do not reach hospital settings), which suggests that the scale of the problem in our geographic region may be larger than currently reported. Therefore, cases associated with the manifestation of delayed effects and chronic toxicity are unlikely to be accounted for by us. The lower number of documented cases involving pediatric patients (who are typically the most frequently affected group in such intoxications) also serves as a critical indicator of the need for heightened vigilance in identifying these incidents [62]. On the other hand, we validate the statement made by Froberg et al. (2007) that more severe poisonings often affect adults who have mistaken the plant for an edible species or have intentionally ingested the raw plant or its extracts in an attempt to experience its perceived medicinal or toxic properties [63]. These observations further reinforce the need for greater public awareness and an in-depth analysis of the issue of plant poisonings.

In contrast to some retrospective studies with a high frequency of exposure to oxalate-containing plants from the Araceae family, anticholinergic, cyanogenic, and ricin syndromes were the most frequently observed clinical effects by us [2,11,21]. This is not unexpected, as it reflects both the geographic distribution of the local flora and the cultural characteristics and traditions of the population. In fact, a review of other literary sources reveals that these factors consistently exert a significant influence on the type and frequency of plant intoxications, with certain populations being particularly affected due to their higher consumption of plant-based food and the associated opportunities for toxin exposure [64,65,66,67,68,69]. In this regard, the plant most frequently responsible for intoxication during the reported period, whether through accidental or intentional use, was *Datura stramonium* L. It is very popular in Bulgaria, including as an ornamental flower, and is known for its hallucinogenic and toxic properties [62,70,71,72]. In the USA, its seeds are used to prepare hallucinogenic teas. One hundred of them contain up to 6 mg of atropine, a dose that can be fatal [29]. Overall, the patients hospitalized in our toxicology department were among those mildly to severely affected by plant poisonings.

It is important to acknowledge the need to differentiate whether each intoxication is due to exposure to plant material or to the phytoproducts that have become increasingly popular in recent years. According to Lüde et al. (2016), the symptoms in both cases may differ significantly, as in the former, exposure is typically to a single plant species, whereas dietary supplements often contain a variety of plant constituents, which complicates the clinical picture [23]. For this reason, in our study, single intoxications with plants, parts of plants, or their extracts were considered, with the presence of intoxication due to complex plant-based dietary supplements set as an exclusion criterion.

The main limitations of the present study are its single-center and retrospective design. In addition, the inclusion/exclusion criteria we established led to a relatively small number of cases. Similar to Fuchs et al. (2011), we believe that the latter is an important condition for properly interpreting the results, as, like many other studies, we were unable to obtain an analytical confirmation of plant toxins in most cases [7]. The present work provides valuable epidemiological data on poisonings in the studied region over a significantly long observation period compared to other retrospective studies. Given that evidence regarding therapeutic procedures in plant poisonings remains generally weak, it seemed important to summarize our experience in the field [5]. The shared therapeutic approaches prevented fatal outcomes in all patients admitted to our medical unit, including those with instances of intentional self-poisoning, where lethality is typically higher [11]. Certainly, the possibility of performing gastric lavage proved to be largely beneficial due to the oral nature of the exposures (with the exception of local cutaneous toxicity), as well as the fact that the poisoning occurred shortly before medical assistance was sought.

Through this work, we aim to raise awareness among healthcare professionals and the public regarding the insufficiently studied issue of plant intoxications. Vigilance regarding the intoxications reported by us should remain high, as some authors have recorded fatal cases following identical plant exposures [7,10,11,12]. Thus, we advocate for greater consistency and increased public visibility in reporting such clinical cases.

## 4. Conclusions

The present study reports the frequency and nature of plant intoxications due to exposure, as documented by the Clinical Toxicology Department at the Naval Hospital in Varna, Bulgaria, for the period 2003–2023. Cases of plant intoxication were relatively rare, with patients admitted to our unit predominantly exhibiting mild to severe symptoms. Anticholinergic syndrome, cyanogenic syndrome, and ricin syndrome were the most frequently observed clinical effects. Diagnosis was made mainly through history, objective examination, and clinical course. In the majority of cases, treatment was based on the observed toxicological syndromes, which led to the prevention of fatal outcomes in all admitted individuals. Although children are generally recognized as the most vulnerable group, only two of the patients were under the age of six. However, we consider that geriatric patients should also be treated with particular caution, as they are prone to using plants for nourishment or self-medication. Further efforts are required to detect and assess cases of individuals who have been toxically exposed to plants but did not seek medical assistance due to the absence or mildness of their symptoms. Preventive measures, such as health education and timely first aid, remain essential for preventing and effectively managing plant poisoning.

## 5. Material and Methods

Through a single-center retrospective review of the data from the Clinical Toxicology Department at the Naval Hospital in Varna, Bulgaria, documented cases of patients with adverse effects related to plant exposure were collected for the period 2003–2023. This hospital unit is the only one in Northeastern Bulgaria specializing in the treatment of acute poisonings and burns, as well as being the only hyperbaric center and intensive cardiology department outside the capital, Sofia. This makes it a strategically important health facility at the national level, with the data it reports being representative of the geographical region.

Each case was thoroughly reviewed and independently assessed by clinical toxicologists at the Clinical Toxicology Department of the Naval Hospital in Varna, Bulgaria, as well as by experts in pharmacology and pharmacognosy from the Medical University “Prof. Dr. Paraskev Stoyanov”, Varna, Bulgaria. All assessments of plant intoxication cases were made by consensus. The diagnoses were based mainly on history, objective examination, and clinical course. The only toxin identified in the clinical laboratory of the hospital was oxalic acid (in urine samples), while the others were not detected in the laboratory. The registered telephone inquiries made about the toxic effects of plants and the submitted expert opinions on court cases related to these cases were analyzed. All the aforementioned actions were carried out using fully anonymized data, with the reviewed information solely being accessible to and processed by the members of the research team.

As in previous similar studies, the severity of the intoxication cases was assessed using the Poisoning Severity Score (Table 3), developed by the European Association of Poison Centres and Clinical Toxicologists, the International Programme on Chemical Safety, and the European Commission [7,23,73,74].

The following criteria were observed for the inclusion or exclusion of individual clinical cases [23,75]:❖**Inclusion criteria:**○Exposure to plants or plant extracts by the patients;○Medical follow-up had to be entirely within the hospital setting, encompassing the entire course of the illness, symptom management, stabilization of the condition, and patient discharge;○Establishment of a causal relationship between plant exposure and the observed clinical effect (determination of a temporal connection between plant exposure and the manifested symptoms; absence of other simultaneous exposures with similar symptoms; and presence of symptoms characteristic of the given toxin or those that are logically consistent from a toxicodynamic perspective).❖**Exclusion criteria:**○Cases of asymptomatic plant exposure;○Simultaneous exposure to other substances with similar symptoms;○Intoxication due to plant-based dietary supplements;○Exposure outside the study period.

The descriptive statistics and graphical representation of the percentage distribution of intoxications with identified toxidromes over the study period were performed using OriginLab^®^ 9.0 software.

## Figures and Tables

**Figure 1 toxins-17-00197-f001:**
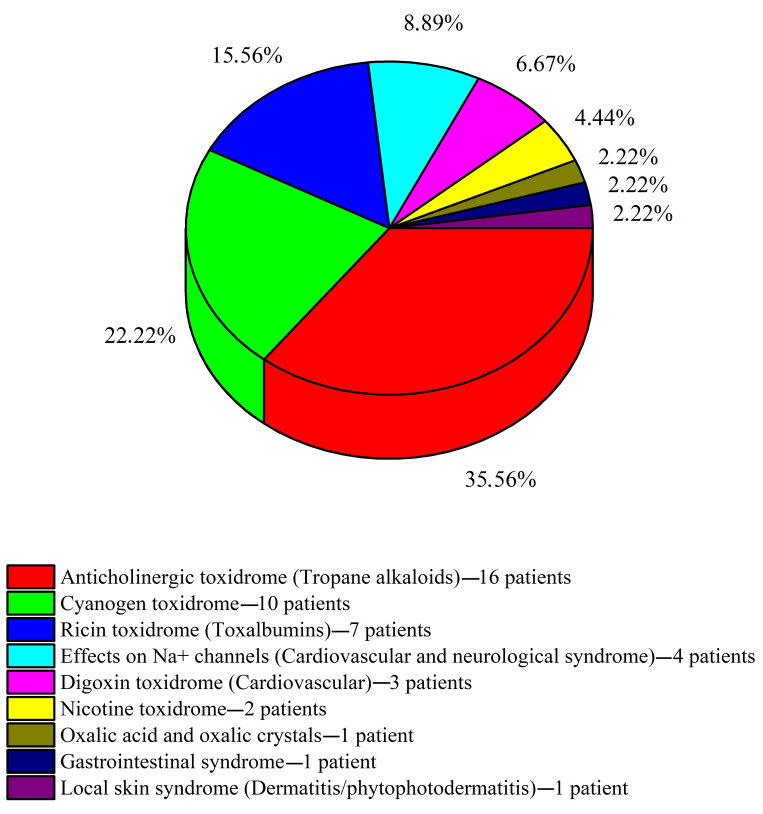
Percentage distribution of intoxications with identified toxidromes (time period: 2003–2023; total patient cohort: 45).

**Table 1 toxins-17-00197-t001:** Plant toxins.

Group	Representatives
Alkaloids	Aconitine, arecoline, Belladonna alkaloids, berberine, lysergic acid amide and ethylamide, lysergic acid derivatives, caffeine, cathinone, colchicine, coniine, emetine/cephaline, ergotamine, N-methylcytisine, pyrrolizidine derivatives, sanguinarine, swainsonine, theophylline, veratridine, vincristine
Cardiac glycosides	Asclepin and related cardenolides, convallatoxin, digitoxin, hederacoside C, α-hederin, hederagenin, hellebrin, oleandrin, strophanthin (~40 derivatives), scillaren A and B, thevetin
Anthraquinone glycosides	Aloinosides, atractyloside, barbaloin, iso-barbaloin, cascarosides, emodin, gummiferine, frangulins, o-glycosides, rhein anthrones, sennosides
Cyanogenic glycosides	Amygdalin, cyacasin, emulsin, glucosinolate, linamarin, progoitrin, ranunculin, protoanemonin, salicin
Proteins, peptides, aminoacids	Abrin, hypoglycin
Lectins	Crucin, ligatoxin, phoratoxin, phytolaccatoxin, ricin, Robinia lectins
Terpenoids	Anasatin, elemicin, hyperforin, gossypol, grayanotoxins, kawain, lantadene A and B, myristicin, phylloerythrin, phorbol esters, pulegone, ptaquiloside, thujone, urushiol oleoresins
Phenolics/ Phenylpropanoids	Asarin, bergamottin, esculoside, capsaicin, coumarin, naringenin, podophyllin, primin, tannic acid, Toxine T-454
Saponins	Glycyrrhizin
Steroid saponins	Aglycones, diosgenin, sarsasapogenin, smilagenin, yamogenin
Carboxylic acids	Asclepin, oxalates, oxalate raphides
Carbohydrates	Psyllium

**Table 2 toxins-17-00197-t002:** Distribution of patients by severity of toxicity.

	None	Minor	Moderate	Severe	Fatal
*Anticholinergic syndrome*	-	-	5	11	-
*Cyanogenic syndrome*	-	7	3	-	-
*R* *icin syndrome*	-	-	1	6	-
*Effect on Na^+^ channels*	-	-	-	4	-
*Cardiovascular syndrome*	-	-	2	1	-
*Nicotine syndrome*	1	-	1	-	-
*Oxalic acid and oxalic crystals*	-	-	-	1	-
*Gastrointestinal syndrome*	-	1	-	-	-
*Local skin syndrome*	-	-	1	-	-

**Table 3 toxins-17-00197-t003:** Poisoning Severity Score.

**None**	No symptoms or signs related to poisoning
**Minor**	Mild, transient, and spontaneously resolving symptoms
**Moderate**	Pronounced or prolonged symptoms
**Severe**	Severe or life-threatening symptoms
**Fatal**	Death

## Data Availability

The original contributions presented in this study are included in the article. Further inquiries can be directed to the corresponding author(s).

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
