# Peer review of "A 20-Year Retrospective Analysis of Plant Poisoning Cases at the Naval Hospital, Varna, Bulgaria"

_toxins, 2025, doi:10.3390/toxins17040197_

Round 1
Reviewer 1 Report
Comments and Suggestions for Authors
This manuscript systematically summarizes the reported cases of plant poisoning in the clinical practice of the Naval Hospital in Bulgaria over a 20-year period. The most commonly observed clinical effects were anticholinergic, cyanogenic, and ricin syndromes. Although some patients presented severe symptoms, no fatality was recorded. This topic is interesting and may generate broad attention in the clinical toxin field. The experiments were well-designed and performed, and the results were presented effectively. Overall, this manuscript is well-prepared, and the findings could significantly benefit the filed. To move the manuscript forward to publishable level, couple of issues need to be addressed:
- The background of ricin biology has some scientific mistakes. For example, there is only one disulfide bound between ricin A and B; ricin B binds to terminal GalNAc or beta-1,4-linked galactose; ricin A targets 28S rRNA in 60S subunit of ribosome. This part needs to be re-wrote and extra references should be added.
- Beside percentages, the number of patients for each toxidrome is suggested to be added in Figure 1.
- The severity of the identified toxidromes in the 45 patients also needs to be summarized.
- The Language can be further polished.
Comments on the Quality of English Language
Can be improved.
Author Response
Comment 1:
This manuscript systematically summarizes the reported cases of plant poisoning in the clinical practice of the Naval Hospital in Bulgaria over a 20-year period. The most commonly observed clinical effects were anticholinergic, cyanogenic, and ricin syndromes. Although some patients presented severe symptoms, no fatality was recorded. This topic is interesting and may generate broad attention in the clinical toxin field. The experiments were well-designed and performed, and the results were presented effectively. Overall, this manuscript is well-prepared, and the findings could significantly benefit the filed. To move the manuscript forward to publishable level, couple of issues need to be addressed:
The background of ricin biology has some scientific mistakes. For example, there is only one disulfide bound between ricin A and B; ricin B binds to terminal GalNAc or beta-1,4-linked galactose; ricin A targets 28S rRNA in 60S subunit of ribosome. This part needs to be re-wrote and extra references should be added.
Beside percentages, the number of patients for each toxidrome is suggested to be added in Figure 1.
The severity of the identified toxidromes in the 45 patients also needs to be summarized.
The Language can be further polished.
Response 1:
All comments have been taken into account, and the necessary corrections have been made accordingly.

Reviewer 2 Report
Comments and Suggestions for Authors
Thank you for the opportunity to review "A 20-year retrospective analysis of plant poisoning cases at the Naval Hospital, Varna, Bulgaria." This a well written review of just 45 cases of plant poisoning. It includes a good introduction into poisoning, diagnosis and treatment of these cases. Certainly this information is useful for the Naval hospital and medical care providers in adjacent Bulgaria. But the authors probably should emphasize that the imply incidence of plant poisoning is much higher in other locations and cultures. Populations and locations have unique customs, diets, diets, herbal uses and plant communities. For example, oleander seeds are commonly used as suicide agents in some cultures. Other populations are dependent on potentially toxic plants for food or their food production system may allow contamination; with dangerous plants toxins. Certainly this manuscript is worthy of publication and will be a good addition to your journal, but the authors might stress the importance of similar local reviews to better identify location and population risks.
Author Response
Comment 1:
Thank you for the opportunity to review "A 20-year retrospective analysis of plant poisoning cases at the Naval Hospital, Varna, Bulgaria." This a well written review of just 45 cases of plant poisoning. It includes a good introduction into poisoning, diagnosis and treatment of these cases. Certainly this information is useful for the Naval hospital and medical care providers in adjacent Bulgaria. But the authors probably should emphasize that the imply incidence of plant poisoning is much higher in other locations and cultures. Populations and locations have unique customs, diets, diets, herbal uses and plant communities. For example, oleander seeds are commonly used as suicide agents in some cultures. Other populations are dependent on potentially toxic plants for food or their food production system may allow contamination; with dangerous plants toxins. Certainly this manuscript is worthy of publication and will be a good addition to your journal, but the authors might stress the importance of similar local reviews to better identify location and population risks.
Response 1:
The influence of the geographic distribution of the local flora, along with the cultural characteristics and traditions of the population, has been discussed and supported with the necessary citations.

Reviewer 3 Report
Comments and Suggestions for Authors
This is an excellent and absorbing manuscript. As a person keenly interested in plant poisonings, I greatly enjoyed reading it. I particularly applaud the inclusion of examples of related toxins and plant species that cause similar toxidromes. I hope that this paper will prompt other hospitals to conduct similar reviews of plant poisoning cases that they have treated.
There appears to be a formatting error in lines 251 to 253. It appears that a change of text was commenced but not completed.
I do not understand the text in line 410. Is "latter" meant, rather than "letter"?
In line 412, I recommend changing "case" to "cases"
Author Response
Comment 1:
This is an excellent and absorbing manuscript. As a person keenly interested in plant poisonings, I greatly enjoyed reading it. I particularly applaud the inclusion of examples of related toxins and plant species that cause similar toxidromes. I hope that this paper will prompt other hospitals to conduct similar reviews of plant poisoning cases that they have treated.
There appears to be a formatting error in lines 251 to 253. It appears that a change of text was commenced but not completed.
I do not understand the text in line 410. Is "latter" meant, rather than "letter"?
In line 412, I recommend changing "case" to "cases"
Response 1:
All identified technical and typographical errors have been corrected in the text.
